# One-pot universal initiation-growth methods from a liquid crystalline block copolymer

Bixin Jin[1,2], Koki Sano [3], Satoshi Aya[3], Yasuhiro Ishida[3], Nathan Gianneschi [4], Yunjun Luo[1,2] & Xiaoyu Li [1,2]

The construction of hierarchical nanostructures with precise morphological and dimensional control has been one of the ultimate goals of contemporary materials science and chemistry, and the emulation of tailor-made nanoscale superstructures realized in the nature, using artificial building blocks, poses outstanding challenges. Herein we report a one-pot strategy to precisely synthesize hierarchical nanostructures through an in-situ initiation-growth process from a liquid crystalline block copolymer. The assembly process, analogous to living chain polymerization, can be triggered by small-molecule, macromolecule or even nanoobject initiators to produce various hierarchical superstructures with highly uniform morphologies and finely tunable dimensions. Because of the high degree of controllability and predictability, this assembly strategy opens the avenue to the design and construction of hierarchical structures with broad utility and accessibility.

---

[1] School of Materials Science and Engineering, Beijing Institute of Technology, Beijing 100081, China. [2] Key Laboratory of High Energy Density Materials, Ministry of Education, Beijing Institute of Technology, Beijing 100081, China. [3] RIKEN Centre for Emergent Matter Science, Saitama 351-0198, Japan. [4] Departments of Chemistry, Materials Science & Engineering, and Biomedical Engineering, Northwestern University, Chicago, IL 60208, USA. Correspondence and requests for materials should be addressed to Y.L. (email: yjluo@bit.edu.cn) or to X.L. (email: xiaoyuli@bit.edu.cn)

The precise synthesis of hierarchical structures, which is ubiquitous in living systems, has captured the imagination of scientists, striving to emulate the intricacy, homogeneity and versatility of the naturally occurring systems, and has represented a central challenge in contemporary materials science and chemistry. In recent years, the fabrication of hierarchical nanostructures with intricate architectures has been demonstrated in a few delicately designed systems, including solid-state aggregation of giant amphiphilic molecules to form long-range ordered bulk morphologies[1], controlled crystallization of nanoparticles to generate superlattices[2], epitaxial growth of crystalline block copolymers to yield exotic nanostructures[3], directional assembly of patchy colloidal nanoparticles to build multidimensional hierarchical superstructures[4], nanoengineering of complicated shapes via DNA origami[5], etc. However, the seeking of a highly efficient and universal method to tailor the assembly of synthetic systems at the nanoscale, simultaneously providing facile and precise control over the hierarchical morphologies and dimensions, remains a very stimulating research topic perpetually.

Meanwhile, the development of supramolecular chemistry, provides an excellent framework for the design and fabrication of a plethora of fascinating artificial structures from molecular machine on the molecular level[6], to supramolecular ploymers[7,8], one- or two-dimensional nanostructures[9,10], and further to macroscopic materials[11]. Regarding the synthesis of uniform nanostructures, most of the successful examples have been achieved through a initiation-growth mechanism, to produce size-controllable nanostructures such as peptide fibers through amphiphilic and hydrogen-bonding interactions[12,13], supramolecular fibers from hydrogen-bonding interactions[14,15], linear supramolecular polymers via shape-promoted intramolecular hydrogen-bonding network[16], functional fibril-like structures from the π-stacking of aromatic molecular amphiphiles[17,18]. In most cases, to suppress self-nucleation and thus to diminish the resultant randomly-sized assemblies, a seeded growth approach is commonly adopted to achieve a decent dimensional control over the assemblies, from which a prior preparation of "seeds" and inevitable multi-step synthesis process are constantly required[19,20]. Very similar approaches were constantly utilized in nature, e.g., to induce the simple fibrilization of amyloid-β and prions[21,22] etc., and also recently applied in crystalline block copolymer systems to produce a fabulous gallery of nanostructures[23–26].

Herein, we report that a previously studied diblock copolymer illustrates striking and unforeseen combination of macromolecular and supramolecular assembly, providing an ideal system for the realization of a initiation-growth strategy, simultaneously producing hierarchical nanostructures with precisely controlled sizes and uniform morphologies. This diblock copolymer P2VP$_{68}$-$b$-PFMA$_{41}$ (P2VP = poly(2-vinyl pyridine), PFMA = poly(2-(perfluorooctyl)ethyl methacrylate), the subscripts represent the numbers of repeating units) forms thermodynamic-favorable cylindrical micelles with a liquid crystalline (LC) micellar core from PFMA block[27–29]. But the relatively quick formation of cylindrical micelles with polydisperse length distribution suggests the considerably low energy barrier for nucleation and fast growth process[30], analogous to one-dimensional supramolecular assemblies[19]. In this work, we discover that by adding a small amount of initiators, which exhibits supramolecular interactions with the copolymer, the subsequent growth driven by the LC ordering effect of the copolymer can yield linear, branched, segmented, hairy plate-like, or star-like nanostructures in a one-pot manner. This in situ initiated assembly approach, able to precisely tune the dimensions of resultant assemblies by carefully adjusting the initiator content

and polymer concentration, presents a straightforward yet highly efficient and universal one-pot synthesis method to produce uniform hierarchical nanostructures.

## Results

**Initiation-growth method for uniform cylindrical micelles.** The first example involves the formation of dynamic covalent bonds with phenylselenyl bromide (PhSeBr) with pyridyl ring[31,32] (Fig. 1a). For an exemplar initiation-growth experiment (Fig. 1b), a 2-propanol (i-PrOH) solution (1.0 mL) of this diblock copolymer (polymer concentration, $C_P = 0.05$ mg/mL) and PhSeBr (percentage of initiator to pyridyl groups, $R_I = 5\%$) was mixed in a glass vial. Subsequently, the solution was held at 80 °C for 20 min and cooled to room temperature (20 °C), whereupon uniform cylindrical micelles are exclusively produced. As can be seen from the transmission electron microscopy (TEM, Fig. 1d) and atomic force microscopy (AFM, Fig. 1e), the initiation section in the middle of these cylindrical micelles appeared thicker than the growth sections at the two ends, resulted from the incorporation of PhSeBr in the initial stage of cylinder formation (Supplementary Figs. 1, 2).

TEM of air-dried samples of the micelle dispersions allowed for visualizing the growth process. Initially at 80 °C, there was no interaction between the PhSeBr and the polymer, and no clear nanostructure was observed (Supplementary Fig. 3a). When the temperature decreased, complexation started to form and discernable aggregates appeared around 70 °C (Supplementary Fig. 3b). The initial aggregates appeared to be amorphous, and only when it was under about 64 °C, the LC ordering can be observed by high-resolution differential scanning calorimetry measurements (Supplementary Fig. 4a). These aggregates, as temperature continues to decrease, evolved into well-defined short rod-like structures (seeds), but with less ordered LC phase, as suggested by our dark field TEM image (Supplementary Fig. 5). Similarly, in situ wide-angle X-ray diffraction measurements (Supplementary Fig. 4b) suggested that the growth of these embryonic seeds into cylindrical micelles (Supplementary Fig. 3c–e) was accompanied by enhanced LC ordering in the micellar core, yielding smectic LC micellar core (as revealed by grazing-incident X-ray diffraction data shown in Supplementary Fig. 4c, d).

The underlying mechanism of this process, termed as initiation-growth process, is depicted in Fig. 1c. Three steps occur spontaneously, namely interaction, initiation and growth step. Firstly, the PhSeBr initiators form complexes with the pyridyl groups on the P2VP chains (interaction step). The resultant less soluble complex tends to aggregate, giving the associated PFMA chains a higher propensity to form oligomeric aggregates with LC cores (initiation step). In the following growth step, these aggregates subsequently serve as seeds for the growth of the free monomeric polymer chains (unimers), driven by the LC ordering effect. Eventually, the fast recruitment of unimers leads to cylindrical micelles with uniform lengths.

We quenched a sample from 80 to 20 °C within 15 s and followed the kinetics via dynamic light scattering intensity (DLS) and TEM (Fig. 1f). TEM results showed that initially the length of the cylindrical micelles increased sharply, then reached a plateau around 5 min, and leveled off after 15 min. Similar conclusion could be drawn from DLS measurements. Assuming the final length of the cylindrical micelles as full conversion of polymers, we can plot the concentration of free polymer chain ([P]) versus time ($t$), which can be fitted into a second order reaction model (Supplementary Fig. 6a), and thus an apparent rate constant ($k$) was determined. By collecting the growth kinetic data at different temperatures (0, 20, and 30 °C), linear relationships were found

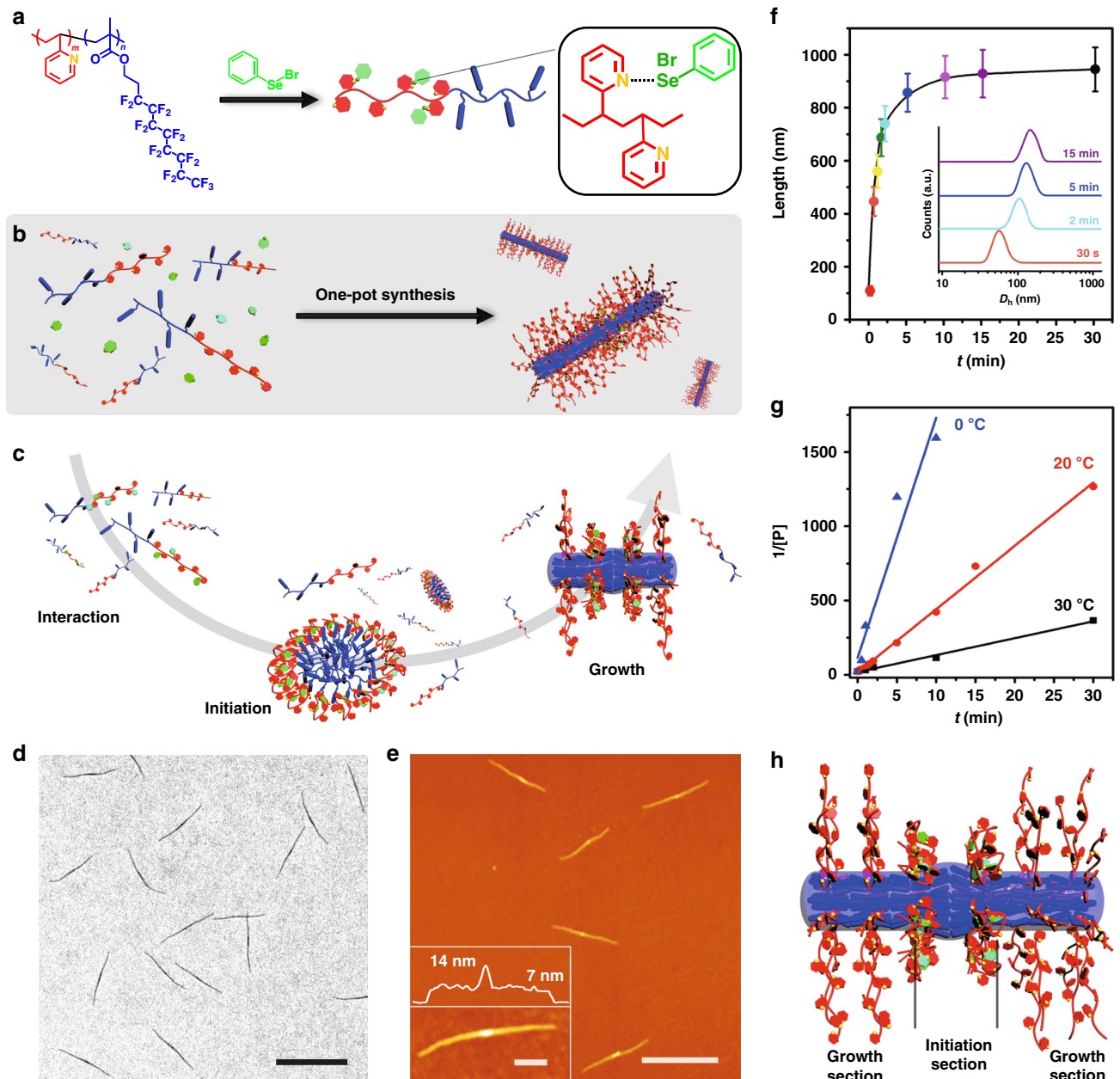

**Fig. 1** Initiated assembly process. **a** Chemical structure of P2VP-*b*-PFMA diblock copolymer and the formation of dynamic covalent bonds between PhSeBr and P2VP units. Schematic illustration of (**b**) the one-pot synthesis of uniform micelles via the initiated assembly process, and the detailed three steps, including (**c**), interaction step, dynamic covalent bond-induced in situ initiation step, and mesogenic ordering induced growth step. **d** TEM and **e** AFM images of the uniform cylindrical micelles from initiation-growth process with $R_I = 5\%$ and $C_P = 0.05$ mg/mL. The variation of micelle length versus time as monitored via TEM (**f**), and the DLS profiles (the inset) from the same sample quenched from 80 to 20 °C in 15 s (error bars represent the s.d.). **g** The linear relationships between 1/[P] and t at different temperatures (0 °C, blue; 20 °C, red; 30 °C, black). **h** Schematic cartoon of a cylindrical micelle produced via this initiation-growth process. Scale bars are 1 μm in the TEM and AFM images, and 200 nm in the inset

between 1/[P] and *t* (Fig. 1g), confirming its second-order nature. Moreover, the descending values of *k* with increasing temperature (Supplementary Fig. 6b) indicates plausibly the higher-temperature dependence of dissociation process[30].

Remarkably, the whole initiation-growth process finished within 15 min, and either quick quenching (within 15 s) or naturally cooling (within 3 h) yielded cylindrical micelles with exactly the same length and markedly narrow polydispersity (typically lower than 1.03; Supplementary Fig. 7 and Supplementary Table 1). The assembly pathways show negligible effect on the final morphology of the micelles, inconsistent with the

preconceived idea that the assembly of block copolymers are generally slow and strongly interfered by the kinetics[33,34]. From this point of view, this process is incoherent with the pathway dependence features from supramolecular systems[19], and also intrinsically distinguished from the previous self-seeding processes, which requires the prior preparation of sacrificial seeds[30,35].

Examination of six different values of $R_I$ and four sets of $C_P$ shows that the length of cylindrical micelles increases with descending values of the two parameters (Fig. 2b). By adjusting the two parameters, the length of cylindrical micelles can be finely

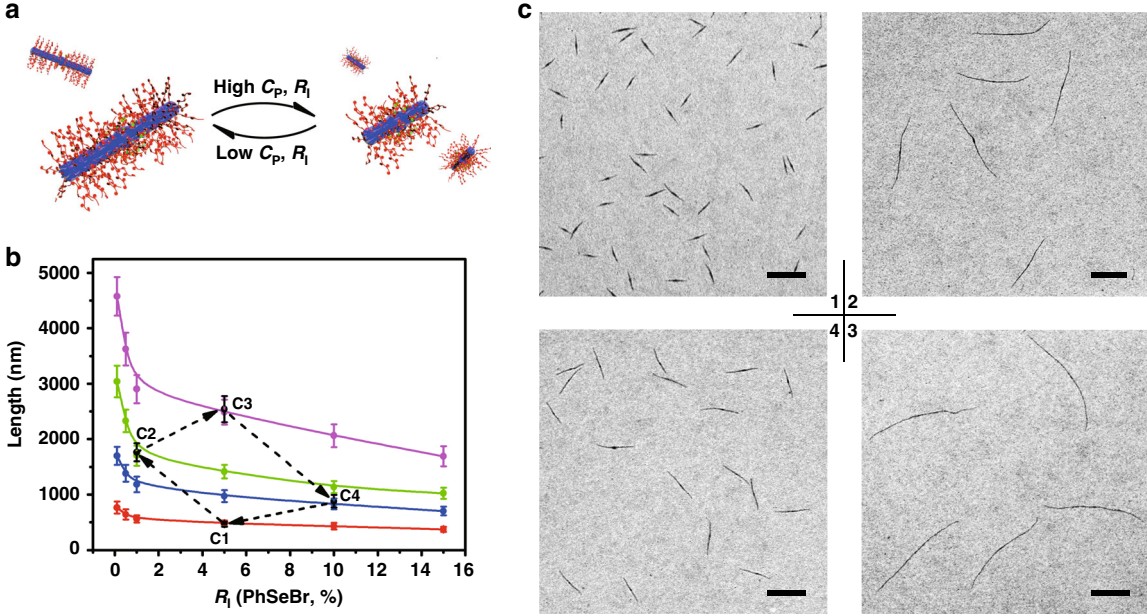

**Fig. 2** The reversible assembly initiated by PhSeBr and length control. **a** Schematic illustrations of the reversible assembly initiated by PhSeBr. **b** The variation of cylindrical micelle length versus $R_I$ with various $C_P$ (red, 0.1 mg/mL; blue, 0.05 mg/mL; green, 0.025 mg/mL; pink, 0.0125 mg/mL; black, transition cycle experiment; error bars represent the s.d.). **c** TEM images of the cylindrical micelles from the transition cycle experiment prepared by heating at 80 °C for 20 min and cooling naturally to 20 °C in 2 h: **C**1, $C_P = 0.1$ mg/mL and $R_I = 5\%$; **C**2, $C_P = 0.025$ mg/mL and $R_I = 1\%$; **C**3, $C_P = 0.0125$ mg/mL and $R_I = 5\%$; **C**4, $C_P = 0.05$ mg/mL and $R_I = 10\%$. Scale bars are 1 μm

tuned from 370 nm ($R_I = 15\%$ and $C_P = 0.1$ mg/mL) up to 4.6 μm ($R_I = 0.1\%$ mol and $C_P = 0.0125$ mg/mL) with fairly narrow length distributions, as implicated by the results from TEM and DLS tests. (Supplementary Figs. 8, 9 and Supplementary Table 2). The growth kinetics of three combinations of $C_P$ and $R_I$ were examined, from which the data can all be nicely fitted into the same one-dimensional growth model (Supplementary Fig. 10a–c). Importantly, the very close values of $k$ from these experiments suggest that it is insensitive to either $C_P$ or $R_I$ (Supplementary Fig. 10d).

Unlike living chain-growth polymerizations, initiation sections of these cylindrical micelles are composed of multiple polymer chains, leading to a varying initiation section length and sometimes slightly larger diameters (Supplementary Table 3). This results in different mass for the initiation section ($M_I$), though the total mass of the micelle ($M$) is mostly determined by the cylinder length. Careful analysis of the nanostructure images reveals a linear relationship between the ratio $M/M_I$ and the inverse of $R_I$ ($1/R_I$) (Supplementary Fig. 11), intimately resembling living chain-growth polymerization or other similar assembly processes[36]. This suggests the effect of $C_P$ on micelle length is originated from the dependence of $M_I$ on $C_P$ (Supplementary Table 3), plausibly attributed to the supramolecular nature of these interactions between the initiator and pyridyl groups. The detailed mechanistic interpretation remains enigmatic and requires deeper investigation beyond the scope of the current manuscript.

These micelles are stable and remain uniform over months of storage (Supplementary Fig. 12). Meanwhile, due to the thermally reversible feature of the dynamic covalent bonds[31,32], when these cylindrical micelles were heated to 80 °C, the dynamic covalent bonds were broken apart, and the LC core was also converted to isotropic state, leading to re-dissolution of the cylindrical micelles to their monomeric state (Supplementary Fig. 13b). Unexpectedly, when the solution was subsequently cooled down to room temperature, the cylindrical micelles were reproduced with

exactly the same length (Supplementary Fig. 13c). Considering the dependence of micelle length on $C_P$ and $R_I$, we designed a transition cycle experiment to demonstrate the micelle lengths are interconvertible (Fig. 2c, and also the black points and arrows in Fig. 2b). By manipulating $C_P$ and $R_I$, the micelle lengths can be varied from 480 nm (point C1), to 1760 nm (point C2), to 2540 nm (point C3), to 880 nm (point C4), and finally back to 480 nm. This transition cycle clearly demonstrates the interconvertibility of these cylindrical micelles (Fig. 2a), distinguishing this assembly approach from chain-growth polymerization or other similar assembly processes[30,35].

**Universal initiation-growth method.** To further show the universality of this assembly method, we examined several other small molecules as initiators, including D-tartaric acid (DTA) and D-lactic acid (DLA) for hydrogen bonds, 1,2,3,4,5-pentafluoro-6-iodobenzene (PFIB) and 1,4-diiodotetrafluorobenzene (DIFB) for halogen bonds, dimethylsulfate (DMS) and benzyl chloride (BC) for quaternization, and copper acetate ($Cu^{2+}$) and Karstedt's catalyst (Pt(0)) for coordination (Fig. 3a). Exactly the same heating-cooling procedures were applied. Despite of the strikingly similar linear morphology in all cases, the choice of initiators can substantially influence the micelle length. The length distribution of the micelles with a fixed $R_I$ (5% mol) and $C_P$ (0.025 mg/mL) were summarized in Fig. 3b (Supplementary Fig. 14 and Supplementary Table 4). The micelle length is determined by the capability of initiators to form complex with P2VP chains. Those initiators binding stronger to the P2VP chains can form more seeds, leading to shorter cylindrical micelles, as implicated by the results from our NMR characterizations at elevated temperatures (Supplementary Fig. 15). Due to the higher sensitivity of spot energy-dispersive X-ray analysis toward metal elements, the existence of Cu and Pt in the initiation section could be easily verified (Supplementary Fig. 16).

The cylindrical micelle lengths from the five different initiators (PhSeBr, DTA, DMS, PFIB, and $Cu^{2+}$, with the same $C_P$)

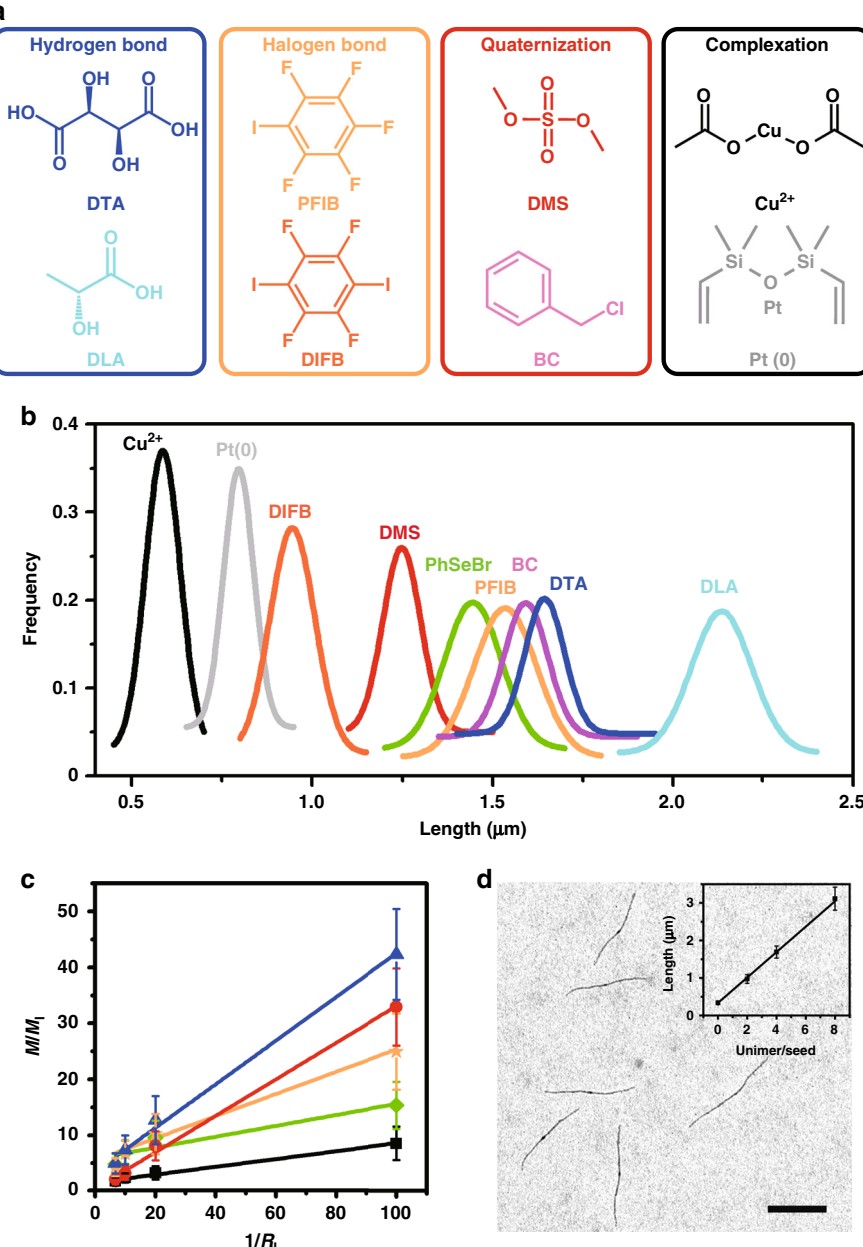

**Fig. 3** The initiated assembly by small-molecule initiators. **a** The chemical structures of all the small-molecule initiators used in this study. **b** The Gaussian distribution of the micelle lengths from the micelles initiated with different initiators at $C_P = 0.025$ mg/mL and $R_I = 5\%$. **c** Linear relationship between $(M/M_I)$ and $(1/R_I)$ for DTA (blue triangle), DMS (red circle), PFIB (orange star), PhSeBr (green diamond), and $Cu^{2+}$ (black square) at $C_P = 0.025$ mg/mL (error bars represent the s.d.). **d** TEM image of the cylindrical micelles produced from the thermo-seeded growth process with a unimer/seed = 4:1. Inset shows the linear relationship between the cylindrical micelle length and the unimers to seed mass ratio. Scale bar is 1 μm

descended with ascending $R_I$ in a parallel manner (Supplementary Figs. 17, 18 and Supplementary Table 5), and in all cases, $M/M_I$ is linearly related to $1/R_I$ (Fig. 3c and Supplementary Table 6). Interestingly, we found for those initiated via hydrogen-bonding and halogen bonding interactions, the micelle formation process was reversible as well, and the micelles length can be varied by changing the $C_P$ and $R_I$. Meanwhile, for those via quaternization and coordination, even with a dilution of four times while retaining $R_I$, subsequent heating-cooling processes could not cause a noticeable change to the micelle length (Supplementary Figs. 19, 20 and Supplementary Table 7), since these complexes were effectively unbreakable upon mild heating. Taking advantage of the high stability of these crosslinked

initiation section from $Cu^{2+}$, thermo-seeded growth of cylindrical micelles can be achieved by adding more polymers and followed by a heating-cooling process. A linear relationship was established from the micelle length and the ratio between the mass of additional polymer and seeding polymer[36] (Fig. 3d, Supplementary Fig. 21 and Supplementary Table 8). On the contrary, if the initiators are removed by adding extra molecules with stronger interactions, randomly-sized cylinders were obtained (Supplementary Figs. 22, 23), suggesting the importance of initiators.

**Fabrication of hierarchical nanostructures.** The third set of efforts was made to investigate how differently macromolecules

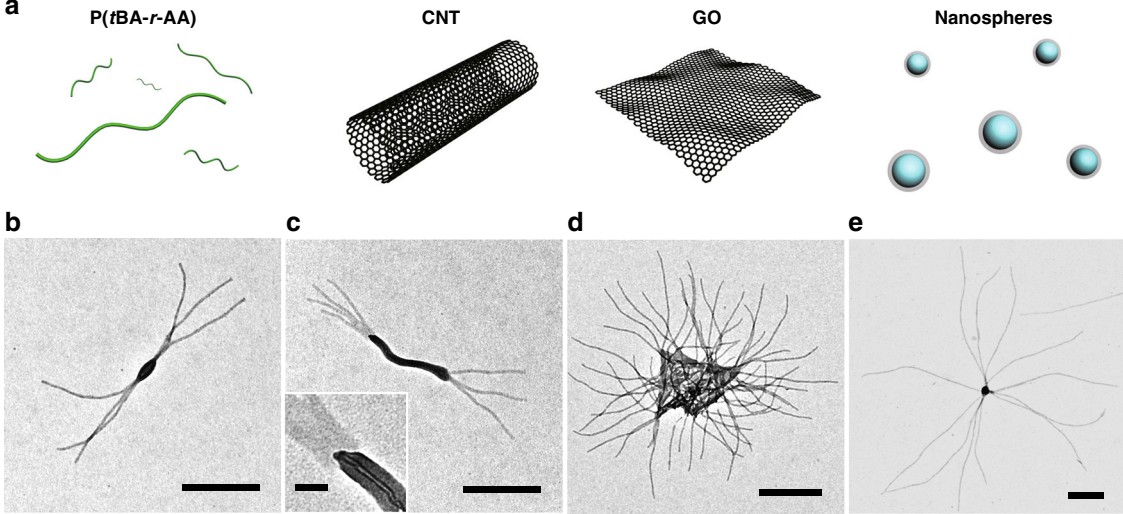

**Fig. 4** The initiated assembly by macromolecular initiators. **a** Schematic cartoons of the different large initiators. **b–e** The TEM images of the hierarchical structures initiated by **b** P(*t*BA-*r*-AA) ($R_I = 2$ wt.%; $C_P = 0.05$ mg/mL); **c** CNTs ($R_I = 10$ wt.%; $C_P = 0.025$ mg/mL); **d** GOs ($R_I = 2$ wt.%; $C_P = 0.05$ mg/mL), and **e** PS nanospheres (NS, $R_I = 2$ wt.%; $C_P = 0.025$ mg/mL). Scale bars are 500 nm, and 50 nm in the inset

would function as initiators (Fig. 4a). Poly((*tert*-butyl acrylate)-*random*-(acrylic acid)), or P(*t*BA-*r*-AA) (hydrolysis degree of 70%, Supplementary Fig. 24), were able to bind with multiple P2VP-*b*-PFMA chains and to initiate the growth of several cylindrical micelles, forming a short and thick rod-like initiation section and eventually a branched structure (Fig. 4b). Considering P(*t*BA-*r*-AA) as zero dimensional objects, we subsequently utilized other nano-objects for the initiation of assembly, such as carbon nanotubes with hydroxyl surface groups (CNTs), graphene oxide nanosheets (GOs), and polystyrene spheres with carboxyl surface groups (NSs, diameter of 90 nm) as one-, two-, and three-dimensional initiators, respectively (Supplementary Fig. 25). The geometry and shape of initiators largely dictated the directions of growing micelles and several hierarchical structures were obtained. From both ends of the CNTs, where the hydroxyl groups are concentrated[37], multiple cylindrical micelles grew and block-like structures were obtained (Fig. 4c, Supplementary Figs. 26b, 27a). Cylindrical micelles were raised from the surface and edges of GOs, giving rise to hairy plate-like nanostructures (Fig. 4d, Supplementary Figs. 26c, 27b). Meanwhile, star-like structures were produced by growing cylindrical arms in all directions from the NSs (Fig. 4e, Supplementary Fig. 26d). Significantly, the number of growing micelles will increase with the dimensions and sizes of initiators, and the lengths of the growing micelles were narrowly distributed (PDI < 1.03, Supplementary Table 9) and tunable (Supplementary Fig. 25) as well.

## Discussion
As highlighted in this paper, we demonstrated a facile one-pot strategy to prepare uniform cylindrical micelles or hierarchical nanostructures from a LC block copolymer via an initiated-growth process. Various supramolecular interactions can be utilized to initiate the assembly, proving the high maneuverability of this method. Particularly, depending on the interactions between initiator and copolymer, the assembly process in some cases can be fully reversible, and thus different micelle lengths are inter-convertible by simply manipulating the initiator to polymer ratio and polymer concentration; and in other cases, the initiation sections can be fully crosslinked to enable thermo-assisted seeded growth process. Furthermore, macromolecules, or nano-objects can also be employed to initiate the assembly to fabricate

branched, segmented, hairy plate-like, or star-like hierarchical structures.

From a general point of view, this facile yet efficient initiated assembly approach presents a simple methodology for self-assembly, simultaneously with high-level of control over the assembled nanostructures. It can be envisioned that this method or mechanism can be extended far beyond this particular polymer, and widely applicable to other analogous block copolymer[38,39] and supramolecular systems[40–43]. In addition, it should neither be restricted to the formation of one-dimensional cylindrical structures, but also two-dimensional platelet-like structures[44–46], or even three-dimensional structures[47]. The mechanisms or rules revealed in this work may shed some light on the process of protein folding[48], and also potential future applications for the resulting assemblies in sensing, nano-electronics, catalysis and biomedicine can be envisaged.

## Methods
**Materials.** *Iso*-propanol (*i*-PrOH, 99.5%, anhydrous), phenylselenyl bromide (PhSeBr, 98%), D-tartaric acid (DTA, 98%), D-lactic acid (DLA, 98%), 1,2,3,4,5-pentafluoro-6-iodobenzene (PFIB, 98%), 1,4-diiodotetrafluorobenzene (DIFB, 98%), dimethylsulfate (DMS, 99%), benzyl chloride (BC, 98%), copper acetate (Cu$^{2+}$, 98%) and Karstedt's catalyst (Pt(0), 2% Pt in xylene) were purchased from Sigma-Aldrich. All other reagents were used as received, unless mentioned elsewhere. The diblock copolymer P2VP$_{68}$-*b*-PFMA$_{41}$ (P2VP = poly(2-vinyl pyridine), PFMA = poly(2-(perfluorooctyl)ethyl methacrylate, the subscripts represent the numbers of repeating units) was synthesized via sequential anionic polymerization[30]. Carbon nanotubes with hydroxyl surface groups (CNTs, multi-walled, the inside diameter of 5–12 nm and the outside diameter of 30–50 nm) was purchased from Sigma-Aldrich. Graphene oxide nanosheets (GOs) was purchased from Tanfeng Tech. Inc. Carboxyl-decorated polystyrene spheres (NSs, diameter of 90 nm) were purchased from BaseLine Chrom. Tech. Research Centre. The CNTs, GOs, and NSs solution (1.0 mg/mL, *i*-PrOH) were dispersed via sonication (50 W sonication processor equipped with a titanium sonotrode) at 0 °C for 15 min before use.

**Transmission electron microscopy (TEM).** Samples for TEM were prepared by drop-casting one drop (~5 μL) of the micellar solution onto a carbon-coated copper grid (Beijing Zhongjingkeyi Technology Co., Ltd, mesh 230). Grids were placed on a piece of filter paper in advance to quickly remove excess solvent in 1 s to prevent further morphological change. Bright field TEM micrographs were obtained on a Hitachi H-7650B microscope operating at 80 kV. No staining was applied for TEM samples unless stated elsewhere. Images were analyzed using the Image-Pro Plus 6.0 software, which is free and available online. For the statistical length analysis, a minimum of 300 micelles were carefully traced by hand to determine their contour length. The number average micelle length ($L_n$) and weight average micelle length ($L_w$) were calculated using Eqs. 1 and 2, from measurements

of the contour lengths ($L_i$) of individual micelles, where $N_i$ is the number of micelles of length $L_i$, and n is the number of micelles examined in each sample. The distribution of micelle lengths is characterized by both $L_w/L_n$ and the standard deviation of the length distribution σ.

$$L_n = \frac{\sum_{i=1}^{n} N_i L_i}{\sum_{i=1}^{n} N_i} \qquad (1)$$

$$L_w = \frac{\sum_{i=1}^{n} N_i L_i^2}{\sum_{i=1}^{n} N_i L_i} \qquad (2)$$

Selective staining of P2VP chains was performed by exposing the sample on carbon-coated copper grid to saturated $RuO_4$ vapor for 30 min in a sealed container. Dark field TEM micrographs were obtained on a Tecnai G2 F20 S-TWIN scanning transmission emission electron microscope. TEM-EDX spot analysis was obtained using a Tecnai G2 F20 S-TWIN scanning transmission emission electron microscope operated at 200 kV equipped with an EDX detector. The spot size was 100 nm in diameter. The samples were prepared by drop-casting one drop (~5 μL) of the micelle solution onto a carbon-coated copper or Au grid.

**Atomic force microscopy (AFM).** AFM experiments were conducted directly on the carbon-coated copper grid used for TEM analysis. AFM images were recorded with a Cypher ES (Asylum Research, CA, USA). The images of the cylindrical micelle were acquired in tapping mode in ambient environment.

**Scanning electron microscopy (SEM).** SEM experiments were conducted directly on the carbon-coated copper grid used for TEM analysis. SEM images were recorded by using a Hitachi SU8020 microscope operating at 30 kV. An ultrathin coating of Au (~5 nm) was deposited via high vacuum evaporation.

**Solution state wide-angle X-ray diffraction (in situ WAXD).** In situ WAXS characterization was carried out at BL45XU in SPring-8 (Hyogo, Japan) with a Rigaku imaging plate area detector model R-AXIS IV++. Scattering vector $q$ ($q = 4\pi\sin\theta/\lambda$; $2\theta$ and $\lambda$ = scattering angle and wavelength of an incident X-ray beam [0.90 Å], respectively) and position of an incident X-ray beam on the detector were calibrated using several orders of layer reflections from silver behenate ($d = 58.380$ Å). The sample-to-detector distance was 470 mm, where recorded scattering/diffraction images were integrated along the Debye-Scherrer ring using Rigaku model R-AXIS Display software, affording a one-dimensional scattering profile. The sample solution was sealed in quartz capillary (diameter ~ 1 mm), and heated to 80 °C for 30 min and gradually cooled down to desired temperatures at an average rate of 1 °C/min.

**X-ray photoelectron spectroscopy (XPS) analysis.** XPS spectra were recorded using PHI Quanteral II SXM equipped with the high-performance Al as mono-chromatic light source.

**Dynamic light scattering (DLS).** DLS experiments were performed using a nano series Malvern zetasizer instrument equipped with a 633 nm red laser. Samples were analyzed in 1 cm glass cuvettes at 20 °C. For the light scattering studies, the refractive index of the block copolymers involved was assumed to be 1.60. The results of DLS studies are reported as apparent hydrodynamic radius ($D_{h, app}$), acknowledging that the particles have been modeled as hard spheres in the experiments conducted.

## Data availability

The datasets generated during and/or analyzed during the current study are available from the corresponding author on reasonable request.

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

## Acknowledgements

X.L. is grateful to the financial support from the National Natural Science Foundation of China (Grant number 21604004) and the 1000 Talents Youth Program. Y.S. acknowledges JST CREST Grant Number JPMJCR17N1, Japan.

## Author contributions

B.J. and X.L. conceived the project and wrote the manuscript with input from Y.L. B.J. and X.L. performed the experiments. N.G. provided useful suggestions on the design and explanations of experiments. K.S. and Y.I. performed in situ XRD experiments, and S.A. performed high-resolution DSC experiment. X.L. supervised the project.

## Additional information

**Competing interests:** The authors declare no competing interests.

