## [Peer Review File · Nature Communications]

Reviewers' comments:

Reviewer #1 (Remarks to the Author):

This paper by Li and coworkers is a brilliant piece of work which I recommend for publication in Nature Communications with enthusiasm. The authors have devised a clever strategy for the self-assembly of micellar structures using an initiation-nucleation-growth mechanism similar to the crystallization-driven self-assembly method pioneered by Manners and Winnik, but significantly more versatile and general.

I predict this paper will be highly cited by scientists working in the areas of soft matter nanoscience and polymer chemistry. Existing systems for the self-assembly of linear micelles with length control typically require the use of very specific polymers, such as poly(ferrocenylsilane) or polythiophene. This work demonstrates that similar architectures can be achieved using liquid crystalline behaviour and really any suitable nucleator, which will substantially broaden the scope of nanomaterials that can be accessed. The supporting information is also impressively thorough and I congratulate the authors on their work.

(My only small comment is that the phrase 'materials and chemistry science' appears several times; this should read 'materials science and chemistry')

Reviewer #2 (Remarks to the Author):

please see my comments in the attached.

Reviewer #3 (Remarks to the Author):

This study is concerned with uniform, reproducible, and tunable self-assembled formation from liquid crystalline block copolymer and various initiator to induce aggregation-induced self-assembling process. The work was carefully done. The authors reach their conclusion based on sufficient evidences. The manuscript is well written. However, such self-assembled nanostructure formation sounds reasonable. This may be regarded as one of the self-assembled processes found in nature and does not cover the essential principles and elements in the field of self-assembled nanostructures. Therefore, I suggested that the manuscript could be suitable for more specialized journals.

Reviewer 1:

This reviewer makes very positive comments about the paper and states, such as: *“The authors have devised a clever strategy for the self-assembly of micellar structures using an initiation-nucleation-growth mechanism similar to the crystallization-driven self-assembly method pioneered by Manners and Winnik, but significantly more versatile and general.”* and that *“I predict this paper will be highly cited by scientists working in the areas of soft matter nanoscience and polymer chemistry.”* and also *“This work demonstrates that similar architectures can be achieved using liquid crystalline behaviour and really any suitable nucleator, which will substantially broaden the scope of nanomaterials that can be accessed. The supporting information is also impressively thorough and I congratulate the authors on their work.”*

Finally, the reviewer recommends publication in *Nature Communications*.

The reviewer makes one suggestion:

Comment 1: *My only small comment is that the phrase ‘materials and chemistry science’ appears several times; this should be ‘materials science and chemistry’.*

Response: We really appreciate that the reviewer pointed this out. We have changed the phrase *‘materials and chemistry science’* to *‘materials science and chemistry’* in the manuscript (on page 1 and 2).

Reviewer 2:

This reviewer makes very positive comments about the paper and states, such as: *“The method developed by the authors is more efficient than previously reported ‘the*

seeded growth' approach because the procedure to prepare 'seeds of micelles' is unnecessary. Instead, a variety of nanostructures can be prepared in a controlled and one-pot manner. Interestingly, the authors further demonstrated macromolecules and the nano-objects (e.g., carbon nanotubes, grapheme oxides and polystyrene nanospheres) which are functionalized by carboxylic acid group can perform as initiators, and produce branched, segmented, hairy plate, and star-like nanostructures."and also "The study reported in this manuscript provides a unique and facile method to create well-ordered hierarchical assemblies with potential in sensings, nanoelectronics and biomedicine." Finally, the reviewer recommends publication in *Nature Communications*.

The reviewer makes five further comments:

Comment 1: *Do the authors have a clear hypothesis on the role of liquid crystallinity when fabricating the self-assembled structures? What will happen if the poly(2-(perfluorooctyl)ethyl methacrylate (PFMA) block is replaced by conventional crystalline block?*

Response: This is an important issue and we thank the reviewer for pointing this out. Firstly, as shown in the current study and our previous report (ref#30, *Angew. Chem. Int. Ed.* **2016**, 55, 11392-11396), the diblock copolymer P2VP₆₈-*b*-PFMA₄₁ formed well-defined cylindrical micelles in *i*-PrOH, though it has a block ratio of ~ 1.7, from which spherical micelles would be expected for amorphous block copolymers. This could be attributed to the dictating liquid crystalline ordering effect from the PFMA block. Secondly, in the current study, our *in-situ* wide-angle X-ray diffraction and high-resolution differential scanning calorimetric measurements (Supplementary Fig. 4) suggested that the growth of the embryonic seeds into cylindrical micelles was accompanied by liquid crystalline ordering in the micellar core. Lastly, the current study was focused on liquid crystalline block copolymer P2VP₆₈-*b*-PFMA₄₁, and we have not tried to replace the PFMA block with a conventional crystalline polymer. Therefore, we are not able to give an exclusive conclusion but can only make a plausible guess. Considering the lower enthalpy associated with the smectic to isotropic transition of PFMA block than the melting enthalpy of most crystalline polymers, two-dimensional platelet-like micelles should be expected, as suggested by A.P. Dove, and R.K. O'Reilly (*ACS Cent. Sci.* **2018**, 4, 63–70).

Comment 2: *Also, the characterization on the liquid crystallinity is not sufficient. For example, authors need to identify types of LC phases (e.g. SmA, SmB etc.) by 2D-WAXD.*

Response: This is an important issue and we thank the reviewer for pointing this out. In our previous research based on exactly the same diblock copolymer P2VP-*b*-PFMA (ref#30, *Angew. Chem. Int. Ed.* **2016**, 55, 11392-11396), our differential scanning calorimetry (DSC) analysis of the diblock copolymer revealed a smectic to isotropic phase transition at 84 °C, close to the literature values from PFMA homopolymers, which was determined to be SmA phase by C.K. Ober and coauthors (*Macromolecules* **1997**, 30, 1906–1914). In that report, we also used small-, medium-, and wide-angle X-ray scattering to characterize the liquid crystalline phase from dried cylindrical micelles. Again, all the scattering signals agreed with the results reported from literature very well (*Macromolecules* **1997**, 30, 1906–1914).

Based on the Reviewer's point, we have also performed an additional 2D GI-XRD test on the dried cylindrical micelles. The data were included in Supplementary Figure 4c and 4d, which agreed with our previous study and literatures.

Supplementary Figure 4. (a) High-resolution DSC traces and (b) temperature-dependent *in-situ* WAXS spectra obtained from the *i*-PrOH solution of P2VP₆₈-*b*-PFMA₄₁ and PhSeBr. (c) GI-XRD and (d) plot of the scattering intensity along equator *versus* the scattering vector q of dried cylindrical micelles (obtained by volatilizing the *i*-PrOH solution of P2VP₆₈-*b*-PFMA₄₁ diblock copolymer initiated by PhSeBr, $C_p = 0.1$ mg/mL, $R_1 = 5$ %). The diffraction signals at 1.98, 3.96, 5.97 nm^{-1} at small angles correspond to (001), (002), (003) lamellar planes, and the (001) distance is the interdistance between the lamellar layers. The diffraction signal at 12.13 nm^{-1} at wide angles corresponds to the intermolecular packing distance perpendicular to the local molecular orientation of mesogenic perfluorooctyl moieties.

We have added an additional sentence in the manuscript on Page 4 as

“... yielding smectic LC micellar core (as revealed by grazing-incident x-ray diffraction data shown in Supplementary Figure 4c and 4d).”

and also discussion on the liquid crystalline nature in the Supplementary Information on page S2-3 as section *“Grazing incident x-ray diffraction measurement (GI-XRD)”* as following,

Grazing incident x-ray diffraction measurement (GI-XRD). In order to further explore the involved molecular packing and phase structure, we conducted additional GI-XRD measurement. GI-XRD analysis was performed at 25 $^{\circ}\text{C}$ using a D8 Discover with GADDS (Bruker A.G.) diffractometer operated at 40 kV and 40 mA with Cu $K\alpha$ radiation ($\lambda = 0.15406$ nm), and the sample-to-detector distance was 10 cm. The GI-XRD image and a plot of the scattering intensity as a function of wave number were included as Supplementary Figure 4c and 4d. The signals at 1.98, 3.96 and 5.97 nm^{-1} at small angles correspond to (001), (002) and

(003) lamellar planes. The (001) distance is the interdistance between the lamellar layers. The diffraction signal at 12.13 nm^{-1} at wide angles corresponds to the intermolecular packing distance perpendicular to the local molecular orientation of mesogenic perfluorooctyl moieties. Since the so-called smectic B type lamellar phase usually has a tight intermolecular packing and forms hexatic ordering within a lamellar plane, a sharp wide-angle signal that overlaps with a broad signal as similar to the one we observed at about 12.13 nm^{-1} would be expected. However, the sharp signal is absent in the present data. Therefore, judging from this perspective, the present liquid crystal phase is more likely a smectic A type lamellar phase, instead of other highly ordered smectic phases. Further detailed analysis could be made by measuring an aligned cylindrical micelle sample in future, which can be effort-consuming and beyond the scope of this manuscript.”

Comment 3: *I assume the self-assembled micelles would be not controlled and probably be randomly-sized if the PhSeBr is not introduced. Can the authors provide such a controlled experiment in the supplementary file?*

Response: We agree with the reviewer. In this work, we discovered that the “initiators” were crucial for the formation of uniform structures. We showed in the manuscript that by adding excess amount of 4-dimethylaminopyridine (DMAP) into the solution of PhSeBr-initiated and uniformly-sized cylindrical micelles, after a heating and cooling cycle, the cylindrical micelles became randomly-sized, due to the removal of PhSeBr initiators (Page 7, line 9-11; Supplementary Figure 22 and 23).

Furthermore, without adding “initiators” initially, the size of the self-assembled micelles was uncontrollable, and randomly-sized micelles were obtained. We have added this result as Supplementary Figure 22 (c) on page S28 accordingly.

Supplementary Figure 22. TEM images of (a) the cylindrical micelles initiated by PhSeBr ($R_I = 5 \%$, $C_P = 0.1 \text{ mg/mL}$); (b) the cylindrical micelles obtained by adding equimolar amount of 4-dimethylaminopyridine (DMAP) and heating to $80 \text{ }^\circ\text{C}$ for 20 min, then naturally cooling to $20 \text{ }^\circ\text{C}$; and (c) the cylindrical micelles formed by dispersing the P2VP₆₈-*b*-PFMA₄₁ diblock copolymer in *i*-PrOH without initiators ($C_P = 0.1 \text{ mg/mL}$). Scale bars are 500 nm.

Comment 4: *Why the particular P2VP₆₈-b-PFMA₄₁ composition is chosen in this study? What would happen in the self-assembly if the composition and molecular weights are changed?*

Response: This diblock copolymer was chosen because it was able to form well-defined cylindrical micelles, which well suited the purpose of this study. There are undergoing systematic studies in our group about the influences of compositions and molecular weights on the assembled micellar morphologies, which will be reported soon. So far, our observation is that the micellar morphologies from liquid crystalline diblock copolymer, due to the formation of liquid crystalline phase, is not so sensitive to the composition and molecular weight as coil-coil diblock copolymers do. However, as limited by the length and scope of this paper, these results were not included.

Comment 5: *What is the origin of LC ordering of the fluorinated block, although it is not consisted of conventional rigid-rod like structure?*

Response: PFMA and its liquid crystalline properties have been intensively explored. Its liquid crystalline nature in bulk state have been reported by C.K. Ober and other researchers (*Macromolecules* **1997**, *30*, 1906–1914; *Macromolecules* **2000**, *33*, 6106–6119; *Prog. Polym. Sci.* **2007**, *32*, 1393–1438). According to these studies, the perfluorooctyl group in PFMA is condered as nanoscale rigid rod-like molecules, with a unique helical conformation, capable of forming a liquid crystalline phase. In addition, the self-assembly of block terpolymers containing PFMA blocks in selective solvent has also been investigated and various micellar morphologies were reported (ref#27-29 in the manuscript).

Reviewer 3:

This reviewer provides some positive statements such as: *“This study in concerned with uniform, reproducible, and tunable self-assembled formation from liquid crystalline block copolymer and various initiator to induce aggregation-induced self-assembling process. The work was carefully done. The authors reach their conclusion based on sufficient evidences. The manuscript is well written.”* However, the reviewer also comments that *“This may be regard as one of the self-assembled processes found in nature and does not cover the essential principles and elements in the field of self-assembled nanostructures.”* and suggested that the manuscript could be suitable for more specialized journals.

Response: We thank the reviewer for the positive comments. In this paper, not only we report an unprecedented discovery of assembly process combining the characteristics of supramolecular polymerization and self-assembly of block copolymers, but also we provide detailed analysis of the thermodynamics and kinetics of the assembly processes. Furthermore, we demonstrated this initiation-growth method is not restricted to one particular system, but is quite versatile that many small molecules can be used as initiators and even hierarchical nanostructures with well-controlled dimensions can be fabricated. We certainly believe the work presented in this paper covers the essential principles and elements in the field of self-assembled nanostructures. Indeed, this is appreciated by Reviewer 1 whom states:

“The authors have devised a clever strategy for the self-assembly of micellar structures using an initiation-nucleation-growth mechanism similar to the crystallization-driven

self-assembly method pioneered by Manners and Winnik, but significantly more versatile and general. ” and that “*This work demonstrates that similar architectures can be achieved using liquid crystalline behaviour and really any suitable nucleator, which will substantially broaden the scope of nanomaterials that can be accessed.*”

We believe that the paper will be of substantial interest to scientists work in the areas of soft matter nanoscience and polymer chemistry. In addition, those interested in the broad areas of block copolymers, liquid crystals, self-assembly, supramolecular chemistry and soft materials science will surely find this study of fundamental interest. We therefore feel that *Nature Communications* is the best platform to present our results to a much broader readership (rather than sending it to a specialized journal).

In addition, as requested by editors, we have revised manuscript accordingly. We changed the title of the paper from “Uniform Hierarchical Structures in One Pot: A Universal Initiation-Growth Method from A Liquid Crystalline Block Copolymer” to “Uniform Hierarchical Structures in One Pot: Universal Initiation-Growth Methods from A Liquid Crystalline Block Copolymer”, since the former one contains 16 words. Moreover, we have added the subheadings in the Results section in the manuscript, and the Data availability and Code availability statement as requested.

We believe that the manuscript has been substantially improved by addressing the reviewers’ critiques, and hope that our responses to their points are satisfactory.

Yours sincerely,

Prof. Xiaoyu Li

School of Material Science and Engineering

Beijing Institute of Technology

REVIEWERS' COMMENTS:

Reviewer #2 (Remarks to the Author):

The authors addressed all the concerns and questions raised by this reviewer. Therefore, I recommend this manuscript to be published in Nature Communications.